# The Role of Gorse (*Ulex parviflorus* Pourr. Scrubs) in a Mediterranean Shrubland Undergoing Climate Change: Approach by Hyperspectral Measurements

**DOI:** 10.3390/plants12040879

**Published:** 2023-02-15

**Authors:** Audrey Marteau, Martin Fourmaux, Jean-Philippe Mevy

**Affiliations:** IMBE-UMR CNRS 7263/IRD 237, Aix Marseille University, Avignon University, 13331 Marseille, France

**Keywords:** field spectrometry, reflectance, shrubland, *Ulex parviflorus*, climate change, plant-plant interactions

## Abstract

The goal of this study was to observe the neighbor effect of Gorse, a plant of the Fabaceae family, on three typical species of Mediterranean shrubland: kermes oak, white Cistus and rosemary. For this purpose, a hyperspectral analysis and the application of vegetation indices (VIs) were carried out. We provide the spectral signature of Gorse, which differs mainly from that of its companion species in the band between 700 and 1350 nm. This supposed Gorse effect was tested in natural conditions and in conditions of forced drought to simulate the effects of the climate change predicted for the Mediterranean Basin. Field spectrometry demonstrated the existence of such interactions between the four species. In control stands, the presence of Gorse significantly modifies the spectral responses of kermes, white Cistus and rosemary, mainly in the near-infrared region (700–1350 nm). Both tri- and tetra-specific plant assemblages also exhibited spectral changes, suggesting an indirect effect of Gorse. Under drought conditions, one-way ANOVA followed by Fisher’s LSD test led us to identify the features involved in plants’ coexistence with Gorse. The *Cistus albidus* reflectance spectrum was clearly increased in the presence of Gorse in rain-exclusion conditions. The application of several VIs allowed us to extract new information on the variation of spectral signatures. Unexpectedly, nitrogen supply by Gorse was not shown, except for Cistus, as shown by the VI NDVI (N) analysis. However, this study proved that Gorse can modify the behavior of its companion species in controls, but also in drought conditions, by increasing their photosynthesis activity (NIRvP) and water content (ratio R975/R900). Gorse therefore appears as a key species in the ecosystem of the Mediterranean shrubland, but its high vulnerability to drought leaves a vacant ecological niche in plant communities. While the spectral reflectance increases linearly with the specific richness in the lack of any disturbance, by contrast, climate aridification imposes a double reciprocal profile. This clearly means that multispecific plant communities cope better with climate change. Nevertheless, knowledge of the underlying mechanisms requires further structural, chemical, and biochemical investigation.

## 1. Introduction

Understanding the processes that govern plant-community structure and composition is one of the major challenges in ecology. *Ulex parviflorus* Pourr. Scrubs (Gorse) belongs to the family Fabaceae, which is widespread in Mediterranean shrublands. As a legume, improving soil-nutritional quality [1,2], it is also described as a fire-prone species, in accordance with the ‘kill thy neighbour’ hypothesis [3,4]. These plant behaviors, in terms of both atmospheric-nitrogen fixation and flammability, are important drivers in terrestrial ecosystems and, thus, play a fundamental role in species evolution and ecosystem dynamics.

The Mediterranean Gorse often co-occurs with other woody shrubs, such as *Cistus albidus*, *Rosmarinus officinalis* and *Quercus coccifera*. The ecosystem of the garrigue confers a whole set of ecosystem services upon the Mediterranean population. Currently, these services are threatened by several anthropogenic activities. In particular, the effects of global climate change on the behaviors key structural species in ecosystems remain challenging. In the Mediterranean region, climate change could result in a decrease in precipitation during each of the seasons, in addition to an increase in temperatures of up to 20% more than the global average [5]. This makes the Mediterranean Basin more vulnerable to climate change [6], especially given that it is considered a biodiversity hotspot. The expected reduction in precipitation is about 30% for the dryer areas at the end of this century [7]. An extended drought period may affect the fitness of co-occurring species in different ways, inducing changes in their abundance, and determining novel community assembly [8]. Therefore, the discrimination and monitoring of the garrigue species is essential for mitigating fire risks and climate aridification as the result of the ecosystemic disturbances.

Several studies have been carried out on the adaptation and the resilience capacity of Mediterranean ecosystems in response to environmental changes [9,10,11]. However, knowledge about the distribution and the ecophysiological characterization of shrublands on an extended spatial scale remains scarce. The application of spectroscopy in ecology has made major advances in recent decades, particularly through remote sensing. This technique makes it possible to observe phenomena on very large scales, which in turn makes it possible to better understand the functioning of ecosystems and to generalize the phenomena observed by ground measurements of functional traits. The resolutions of sensors and spectral imagers are becoming increasingly fine, which makes it possible to obtain extremely precise results. The coupling of spectral properties and functional traits is therefore essential for the mapping and spatial–temporal dynamics of the specific compositions of forests from space. Hyperspectral spectrometry is a relevant approach, since it dispenses with taking measurements of plants in destructive ways [12]. Indeed, each plant species has its own spectral signature, which corresponds to the proportion of reflected and re-emitted incident light (reflectance) at each wavelength tested (visible, infrared, and short-wave infrared). These unique spectral bands are due to the leaves’ biochemical and biophysical properties, such as epidermal hairs or wax layers [13].

Recent studies indicated that field spectrometry may be applied in the taxonomic discrimination of plants [14] and even at the level of the characterization of the varieties [15]. Because phenotypes are the result of metabolic differentiation, it is likely that field spectrometry could reveal the fine functional traits of plants in response to their environmental biotic and abiotic constraints. Indeed, intra-specific changes in morpho-physiological traits were detected by UAV imagery from 18 populations of *Pinus nigra* [11].

The goal of this study was to determine optically the impact of a Mediterranean Gorse on three other co-occurring species (white Cistus, kermes oak and rosemary) in the context of climatic aridification. To this end, we examined the following topics: (i) whether the interactions between Gorse and its companion species can be characterized spectrally; (ii) the extent to which Gorse can mediate the water balance of neighboring species; (iii) how Gorse can mediate the nitrogen metabolism and photosynthesis of its companion species; (iv) and which vegetation indices can best reveal the interactions between Gorse and its companion species.

## 2. Materials and Methods

### 2.1. Study Area

The study was conducted at the CLIMED site, an observatory in a shrubby area north of the city of Marseille in the Massif de l’Etoile (43°21′54″ N, 5°25′30″ E). The CLIMED study program simulates the decrease in precipitation of about 30% predicted for the Mediterranean region. The site is mainly composed of 4 shrubby plant species: *Rosmarinus officinalis*, *Cistus albidus* (white rockrose), *Quercus coccifera* (kermes oak) and *Ulex parviflorus* (Gorse of Provence). Two types of device are implanted in plots of 16 m². Rain-exclusion devices (Figure 1A) equipped with gutters and control devices with inverted gutters are used to consider the shading effect (Figure 1B). These devices are installed on plots with different natural combinations of species; a total of 93 plots are arranged on the CLIMED site.

For this study, several types of plant combination were considered (11 assemblies). Monospecific plots for the 4 dominant woody species: *Ulex parviflorus* (Up), *Cistus albidus* (Ca), *Quercus coccifera* (Qc), *Rosmarinus officinalis* (Ro). Bispecific plots with the following associations: UpCa, UpQc, UpRo. Trispecific plots with the following plant combinations: QcCaUp, QcRoUp, CaRoUp, CaQcUp, RoQcUp, RoCaUp. Tetraspecific plots: QcCaRoUp, CaQcRoUp and RoQcCaUp. The first name of species of each assemblage corresponded to the species that was considered for spectral measurements. Each modality of water treatment (control and exclusion devices) was repeated 3 times. Thus, a total of 66 plots were selected to test the neighbor effects of *U. parviflorus*.

### 2.2. Selection of Individuals

Plants were chosen central to the plot to avoid border effect. The selected individuals were located at least 1 m from the edge of the plot. In each plot, 3 individuals were carefully selected; they were located as close as possible to the other species for maximum interaction.

### 2.3. Spectral Measurements

The measurements were performed during a field campaign (27 April 2022 to 10 May 2022). The reflectance was obtained between 350 and 2500 nm using a portable spectroradiometer (FieldSpec^®^ 4 ASD, USA) with a field of view of 25°. First, a reflectance calibration was performed using a Spectralon^®^ disk before plant measurement. For the measurements, the optical fiber was positioned at nadir, 10 cm above the canopy of target plant. On each of the 3 individuals per plot, 3 reflectance measurements were performed. To check for sensor stability, each measurement was repeated 3 times. The data acquisition was performed between 11:00 and 15:00 in similar synoptic conditions. A total of 1215 data were recorded: monospecific (189); bispecific (324); trispecific (486); and tetraspecific (216).

### 2.4. Spectral Post-Treatment

With ViewSpecPro, spectra were corrected by applying a “splice correction” for linear interpolation at the spectral discontinuities. On all data obtained, artifacts representing absorption bands of atmospheric water were observed. Therefore, the bands from 1340 nm to 1430, from 1810 nm to 1980 nm and above 2400 nm were removed for all the data.

### 2.5. Spectral Indices

Vegetation Indices (VIs) are widely used to predict biophysical properties of natural ecosystems. Their principal advantage is that they enhance spectral-reflectance information by detecting the spectral variability that might occur, for instance, in plant canopies or physiological and structural properties of leaves [16]. First, we considered two VIs that are related to plant-water content and photosynthesis.

Water-content index was calculated as follows [17]. R975/R900. NIRvP: structural proxy for SIF (solar-induced fluorescence). Photosynthesis [18]: NIRvP = NIRv × PAR, where PAR is the photosynthetically active radiation. The NIR is the average reflectance between 770 and 780 nm. NDVI $=\frac{NIR-RED\;ref}{NIR+RED\;ref}$, where RED ref is the average reflectance between 650 and 660 nm. The PAR were collected from the meteorological station of the CLIMED site at each time of canopy-reflectance measurement. Second, other VIs were selected to assess plant nitrogen content, as well as their structure and functioning either directly or indirectly linked to photosynthesis. These are recorded in Table 1.

### 2.6. Statistical Analysis

Spectral data were processed from the Metaboanalyst platform [31]. They were first centered and reduced, after which they were normalized for multivariate analysis (partial least-square discriminant analysis (PLSDA)), followed by ANOVA and Fisher’s LSD post hoc tests. The latter allows pairwise comparison of assemblages for each wavelength of spectrum. In the same way, spectral indices were processed by PLSDA and then classified by the VIP scores (importance of the variables in projection on axis 1 of PLSDA). The most discriminatory indices that fitted the objectives of the study were processed by two-way-ANOVA with Statgraphics. Correlation models between total reflectance and species richness were also created with Statgraphics.

### 2.7. Scanning Electronic Microscopy (SEM) Observations

Three twigs of *Ulex parviflorus*, (3 individuals) at different phenological stages (young and mature) were observed with a Zeiss scanning electron microscope, SEM EVO 15 (with lanthanum boride source), in wet mode.

## 3. Results

### 3.1. Spectral Signature of Ulex Parviflorus

The PLSDA analysis exhibited a total inertia of 37% in accordance with a weak discrimination between Gorse, kermes oak and rosemary (Figure 2). However, *U. parviflorus* seems spectrally closer to kermes than to rosemary and cistus. Exploring the spectral profiles showed differences in some areas of the spectrum (Figure 3). The Gorse sems to have exhibited much lower reflectance than the other species in the band between 700 and 1400 nm. This was confirmed by the one-way-ANOVA, followed by the Fisher’s LSD test, with the identification of the wavelengths that significantly distinguished the spectrum of *U. parviflorus* from the 3 other species (Appendix A).

### 3.2. Hyperspectral Response of Interspecific Assemblages

The spectral data obtained according to the different types of assemblage tested under natural conditions (control modality) showed that the bands from 700 to 1400 nm were the most discriminatory (Figure 4). The direct effect of Gorse on its companion species may be observed by comparing the spectra obtained in monospecific and bispecific stands. The pairwise comparison of the assemblages for each wavelength resulted in more than one thousand significant differences (*p* < 0.05). We present in this report the most significant differences obtained for each species (Figure 5). The Gorse did not directly affect the reflectance of Ca at 1030 nm (Figure 5a), nor that of Ro at 559 nm (Figure 5b). However, an increase in reflectance occurred for QcUp compared to Qc at 1291 nm (Figure 5c). Conversely, all the three species (UpCa; UpRo; UpQc) contributed to the increase in the reflectance of Gorse at 2254 nm compared to their monospecific stand (Figure 5d). By plotting the integrated reflectance with the specific richness, we found a positive and linear correlation, meaning that the spectral reflectance increased with species richness (Appendix A).

### 3.3. Hyperspectral Response to Climate Aridification

Figure 6 compares the reflectance of each assemblage as a function of the control and exclusion treatment modalities. The features identified by the one-way ANOVA followed by the Fisher’s LSD test are summarized in Appendix A. In general, there was no significant effect of rainfall exclusion on the monospecific stands (Figure 6a,e,i, Appendix A). In the bispecific stands, there was no significant effect of Up on Ro (RoUp); this effect was only observed in the SWIR2 band (1801–2350 nm). No difference was found for QcUp all over the spectrum (Appendix A). The most significant effects were observed between eCaUp and cCaUp (Appendix A), where the reflectance was 12% higher than the cCaUp at its maximum value, about 1140 nm (Figure 6f). Up appears to play a role in the reflectance capacity of Ca under drought conditions. For the trispecific combination, the lowest reflectance was recorded in the exclusion, especially from the eCaQcUp and eRoQcUP stands (Figure 6g,k). This decrease in reflectance was also observed in the tetraspecific eRoQcCaUP and eQcCaRoUp stands. For the effect of the three species on Up, we noted an inverse response for the tetraspecific assemblage (Figure 6p). In general, the reflectance in the exclusion decreased with the species richness. This property did not follow a linear pattern, but a double inverse profile (Appendix A).

### 3.4. Analyses of SPECTRAL indices of Vegetation

#### 3.4.1. Water-Content Index: The Ratio R975/R900

High reflectance in water absorption bands is associated with low water content. Therefore, the lower the R975/R900 ratio, the higher the water content of the Gorse companion plant. The two-way ANOVA showed a significant effect of the assemblage and water treatment. An interaction remained between these two factors (F = 5.709; *p* < 0.001). We thus compared the two water treatments (control and exclusion) for each assemblage. For the three species in the monospecific stand, the decrease in precipitation had no significant effects on the water contents compared to the controls (Figure 7). In the bispecific stands, Up improved the water-conserving abilities of Qc, while the opposite effect was observed for Ca. The Ro water contents were not affected by Up. However, when Qc was in a tri-specific stand with Ro or Ca, Up no longer improved the water content of Qc, nor did it even reduce it (QcRoUp). The water content of Ro in the trispecific stand was improved in the presence of Ca but reduced in the presence of Qc. In the tetraspecific stand, each of the three species had a better water capacity.

#### 3.4.2. Chlorophyll-a Fluorescence and Photosynthesis Index: NIRvP

We observed a significant interaction between the factor’s assemblage and water treatment for the quantitative variable NIRvP: two-way-ANOVA (F = 11.123; *p* < 0.001). For each assemblage, the control and exclusion were thus compared (Figure 8). In the bispecific stand, Up enhanced photosynthesis in Ca as well as Ro under rain-exclusion devices. This could mean that Up increased the biomass of Ca and Ro under precipitation-exclusion conditions. Furthermore, photosynthesis was not improved when Qc was associated with Up. Up therefore does not influence the biomass production in Qc under drought conditions. However, in the trispecific stand with Ca, Qc and Ro displayed an increase in photosynthetic capacity with rain exclusion.

#### 3.4.3. Nitrogen-Content Index and Other Vegetation Indices

The PLSDA analysis performed on all the selected indices (18) exhibited 4 hits with VIP scores >1 (Appendix A): NPQI, NDVI(N), PRI and SIFb. The two-way ANOVA carried out with the nitrogen-content index, NDVI(N), showed a significant interaction between the assemblage and the water availability (F= 6.921, *p* > 0.001). For each assemblage, the nitrogen levels were compared between the control and exclusion treatments. In the monospecific assemblages (Figure 9), the decrease in precipitation did not have a significant effect on the nitrogen content compared to the controls. However, in the bispecific assemblage, Up improved only the nitrogen content of Ca; in addition, it probably did so in the trispecific (CaQcUp) and tetraspecific (CaQcRoUp) assemblages. It should be noted that the nitrogen content was also increased under the drought conditions for QcCaUp and RoQcUp.

#### 3.4.4. Morphology and Anatomy of *Ulex parviflorus*

The SEM pictures of the *Ulex parviflorus* spines show that these were glabrous, with a high density of oval stomata (Figure 10A,C). The stem shape was fluted, with abundant forked trichomes in pairs, but with a lower density of stomata compared to the spines (Figure 10B,D).

## 4. Discussion

We provided a Gorse spectral signature, which may be distinguished from those of *R. officinalis*, *Q. coccifera* and *C. albidus* in terms of its architectural, structural and chemical organization. Our studies demonstrated how field spectrometry can discriminate between coexisting plant species. We showed that the spectral signature of Gorse differed from that of its companion species in specific spectral bands (Figure 3). The lower overall reflectance of the Gorse may be attributed to its spiny structure. However, the most discriminatory wavelength in the visible domain was exhibited by Rosmarinus and its assemblages (Figure 5). The discrimination of Rosmarinus coexisting with Gorse and the two other species occurred mainly in green reflectance (559 nm). This clearly indicates that Rosmarinus’ neighboring plants affect its chlorophyll-light-absorbing properties and, thus, photosynthesis. Our work agrees with other studies showing that hyperspectral measurements are applicable for taxonomic purposes [14]. This raises the question of the complex entanglement of the structural and/or molecular components involved in species distinction. We also showed that in natural conditions (control), the reflectance of individuals increases linearly and positively with the species richness. This is in accordance with a previous study carried out in the vicinity of the CLIMED site, which showed a linear increase in the plant productivity with species richness [32]. The occurrence of several kinds of species in the same area allows the exploration of different ecological niches, thus limiting interspecific competition. Indeed, the different root-distribution patterns of different species from the same community sharing the same ecological conditions is crucial for the understanding of how plants cope with their environmental constraints. The *Q. coccifera* roots and rhizomes were mainly located in the uppermost 15 to 35 cm of the soils [33], while the Gorse exhibited deep roots, with less lateral-root development in the upper layers [34]. In this study, both the kermes oak and the Gorse clearly explored different ecological niches. Therefore, the question is raised as to what would happen with a very high species richness. The linear profile observed would probably reach a plateau stage in accordance with the hypothesis of functional-trait redundancy.

In the context of climate aridification, the presence of Gorse also results in a modification of the spectral signature of its companion species, in this case, Cistus, rosemary and kermes oak (Figure 6). However, after about ten years of excluding precipitation at the CLIMED site, we observed a total mortality of Gorse in monospecific stands under the exclusion devices. This clearly indicates a change in the landscape of the garrigue due to the ecological niches left vacant by the decline in the monospecific stands of Gorse, which raises many concerns. On one hand, the question of the species that would be most likely to occupy these new niches and, on the other hand, the mechanisms involved in the Gorse’s decline. To address the latter question, the Gorse spines were analyzed through SEM images (Figure 10), which revealed a superficial arrangement of the stomata, with high density and devoid of protective appendages. This vulnerability to the increased atmospheric water demand in drought conditions would explain the great competition between Gorse individuals for the satisfaction of water needs. Indeed, *Q. coccifera* stomata are mainly located on the abaxial side, while *R. officinalis* and *C. albidus* have a high density of trichomes protecting their leaves from desiccation [17]. Gorse therefore persists in species-diverse environments, probably through niche diversity, which results in changes in the spectral response of its companion species. The most significant direct effect was observed with the Cistus (CaUp), whose reflectance in rain-exclusion conditions increased over the entire spectrum (Figure 6f; Appendix A). A higher reflectance means fewer internal-molecular-absorption features, as well as the involvement of structural aspects of leaf-like trichome abundance, wax thickness and texture [35]. This raises the question of the identification of the different molecules that may be characterized in such plant–plant–microorganism interactions. Indeed, we identified wavelengths that discriminated the effect of drought and coexistence with Gorse (Appendix A). Each of these values could correspond to the radiation-absorption properties of specific compounds, leaf structures and canopy architectures. Recent studies have begun to identify the molecular intersections between biotic- and abiotic-stress responses through the concept of plant immunity [36]. Considering the overall richness of the communities investigated, we showed that drought induces a reduction in reflectance with the increase in the specific richness that follows a double reciprocal profile. Our data are in accordance with those of previous studies, which concluded that neighbors increase the profiles of local adaptation, with adaptation to competition occurring in benign habitats [37]. Further studies are required to characterize the role of niche diversity versus that of chemical metabolic changes.

In the specific case of the index linked to the water content of the tissues (ratio R975/R900), the effect of the Gorse varied according to the neighboring species. Gorse seems to exert a positive influence on kermes oak through its ability to conserve water in drought conditions and, therefore, has a facilitating role. This effect turns into antagonism in the case of Cistus, since it has a lower water content in the presence of Gorse in drought conditions. Because Gorse and kermes oak have different rooting systems, the observed facilitation effect is the result of niche diversity rather than the involvement of chemical mediators. However, the intense competition between Cistus and Gorse suggests that Gorse may act through the regulation of oak stomatal opening due either to volatile signal transducers or by the rhizosphere’s chemical components. For instance, plants or rhizospheric microorganisms’ exudation of abscisic acid is known to regulate stomatal conductance [38,39]. Considering all the tretraspecific assemblages, a decrease in the ratio of R975/R900 was noticed, confirming the same pattern as that obtained from the trispecific plant associations [17].

In the case of the role of Gorse as a Fabaceae for nitrogen supply in the ecosystem, a clear improvement was shown for Cistus in the bispecific, trispecific and tetraspecific stands. Although it is recognized as a releaser of allelopathic compounds, these do not affect the soil-nitrogen net rates of *C. albidus* [40]. Improvements were not observed in the bispecific assemblages of *Q. coccifera* and *R. officinalis* (Figure 9). For the rosemary and the kermes oak, the presence of nitrogen was increased in the trispecific stands, RoQcUp and QcCaUp, respectively. Our data raise the question of the nitrogen demand of wild species. Therefore, it is likely that Cistus’ nitrogen metabolism is higher than that of the two other companion species. Nitrogen, as a macronutrient, is a part not only of the composition of proteins, but also of the structure of chlorophyll, which is a major pigment in photosynthesis [41]. Our work attempted to determine the possible variations in the photosynthetic capacity of species through the NIRvP index (Figure 8). The *U. parviflorus* has a significant positive effect on the photosynthetic capacity of *C. albidus.* This is quite in accordance with the improvements in nitrogen content by the neighbor effect of Gorse, but also with other research reports [42,43]. A similar photosynthesis improvement was also obtained with *Q. coccifera* associated with *C. albidus* and Gorse. The *R. officinalis* NIRvP index increased when co-existing with the Gorse, but also in association with *C. albidus*. This result appears to have been dependent on the availability of water rather than nitrogen. In addition, the higher the number of species increases in the assemblages, the greater the photosynthetic capacity seemed, whether in natural conditions or in rain-exclusion conditions. Since NIRvP is positively correlated with gross primary productivity (GPP) and sun-induced fluorescence (SIF), our results indicate that NIRvP is a potential index to estimate the productivity of shrubland ecosystems in their specific diversity from space. Indeed, SIF is one of the major indicators for measuring the physiology of photosynthesis with remote-sensing tools [44]. In addition, our work also identified other potential indicators of ecosystem structure and function, such as NPQI, PRI and SIFb.

## 5. Conclusions

Hyperspectral spectrometry has proven to be a promising nondestructive method for observing the neighbor effect in plant–plant interactions. Indeed, through the characterization of the spectral signature of Gorse, we followed the spatial dynamics of a shrubland following disturbances such as fires using remote-sensing devices. The spectroscopy also demonstrated that in rain-exclusion conditions, Gorse modifies the water content and the biomass of the three other companion species through photosynthesis, depending on their different assemblages. However, with the exception of Cistus, it does not modify nitrogen supply, as previously expected, even though it is a Fabaceae. Among the vegetation indices investigated, NPQI (normalized phaeophytinization index) appears as a robust canopy indicator of Gorse’s neighbor effect. Spectroscopic measurements therefore make it possible to acquire valuable information on the physiologies of ecosystems independently of traditional field and laboratory surveys. This work also highlighted the vulnerability of Gorse to climate change, which urgently requires appropriate conservation measures. Therefore, a better understanding of biotic interactions seems necessary, particularly through the characterization of the metabolic networks underlying the observed spectral signatures.

## Figures and Tables

**Figure 1 plants-12-00879-f001:**
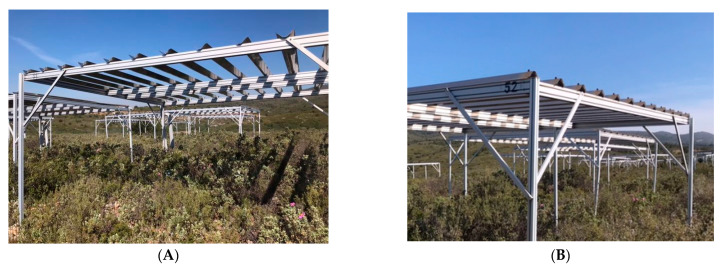
The CLIMED site with rain-exclusion (**A**) and control (**B**) devices.

**Figure 2 plants-12-00879-f002:**
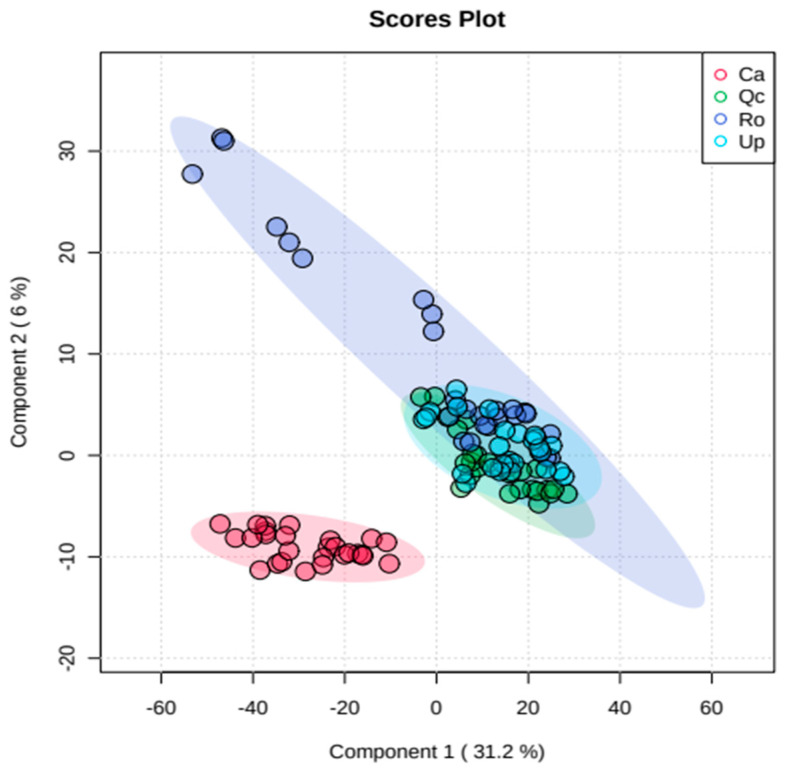
Partial least-squares discriminant analysis (PLSDA) of reflectance spectra of *Ulex parviflorus* (Up), *Cistus albidus* (Ca), *Rosmarinus officinalis* (Ro) and *Quercus coccifera* (Qc) in monospecific stands in control. *n* = 27. Measurements carried out between 27 April and 10 May 2022.

**Figure 3 plants-12-00879-f003:**
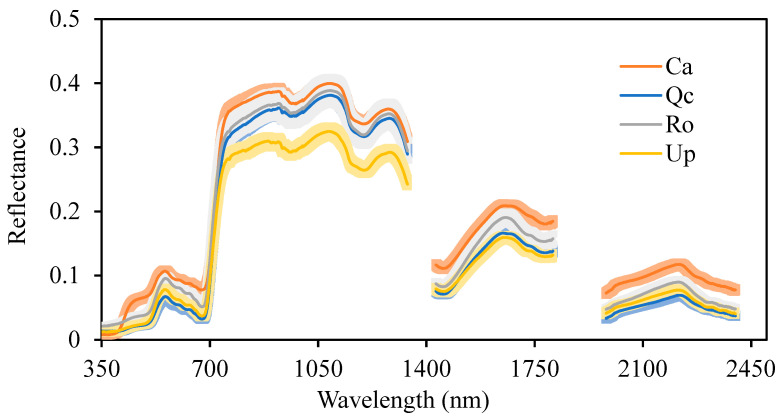
Spectral signatures of *Ulex parviflorus* (UP) *Cistus albidus* (Ca), *Quercus coccifera* (Qc) and *Rosmarinus officinalis* (Ro) in monospecific control stands. Means, *n* = 27 with standard errors. Measurements were carried out between 27 April and 10 May 2022.

**Figure 4 plants-12-00879-f004:**
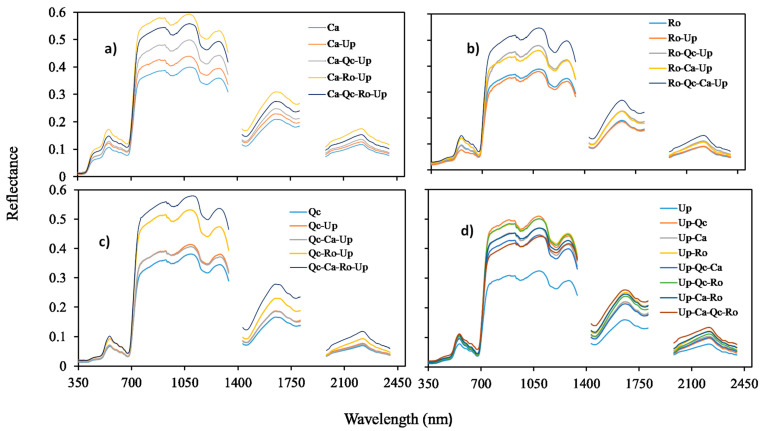
Reflectance spectra in control. (**a**) *Cistus albidus* (Ca); (**b**) *Rosmarinus officinalis* (Ro); (**c**) *Quercus coccifera* (Qc); (**d**) *Ulex Parviflorus* (Up) in monospecific stands. Bispecific (Ca-Up; Ro-Up; Qc-Up; Up-Ca; Up-Ro; Up-Qc), trispecific (Ca-Qc-Up; Ca-Ro-Up; Qc-Ca-Up; Qc-Ro-Up; Ro-Qc-Up; Ro-Ca-Up; Up-Qc-Ca; Up-Qc-Ro; Up-Ca-Ro) and tetraspecific stand (Ca-Qc-Ro-Up; Qc-Ca-Ro-Up; Ro-Qc-Ca-Up; Up-Qc-Ca-Ro). The first two letters of each assemblage indicate the species on which the measurements were performed. Means, *n* = 27.

**Figure 5 plants-12-00879-f005:**
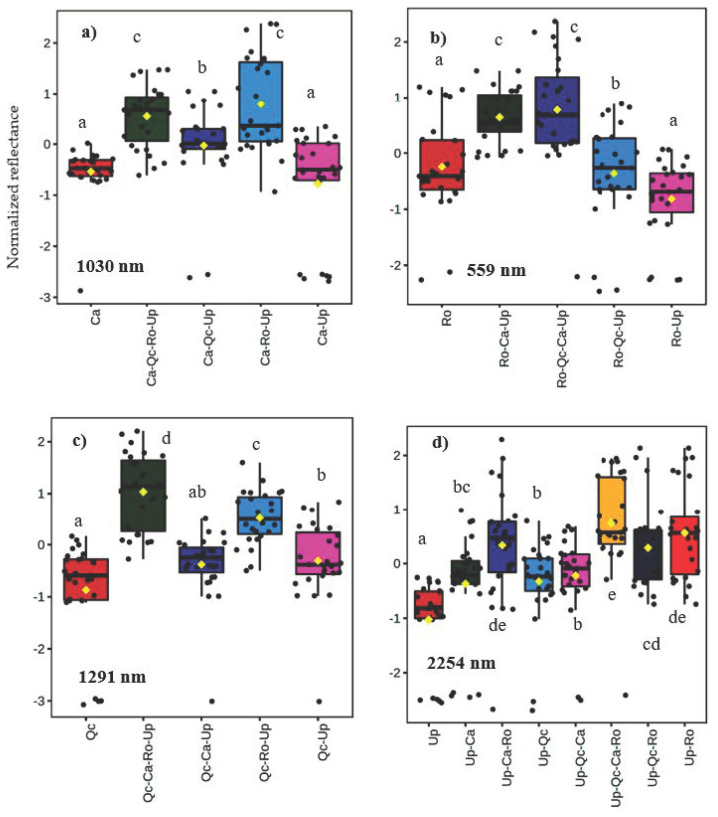
Box plots of the normalized reflectance for the most discriminatory wavelength between the assemblages in the control condition. (**a**) *Cistus albidus* (Ca); (**b**) *Rosmarinus officinalis* (Ro); (**c**) *Quercus coccifera* (Qc); (**d**) *Ulex parviflorus* (Up). The first two letters of each assemblage indicate the species on which the measurements were performed. *n* = 27. ANOVA (*p* < 0.001) followed by LSD test for pairwise comparisons of the different plant associations. Significant differences are shown by different letters: a < b < c < d < e.

**Figure 6 plants-12-00879-f006:**
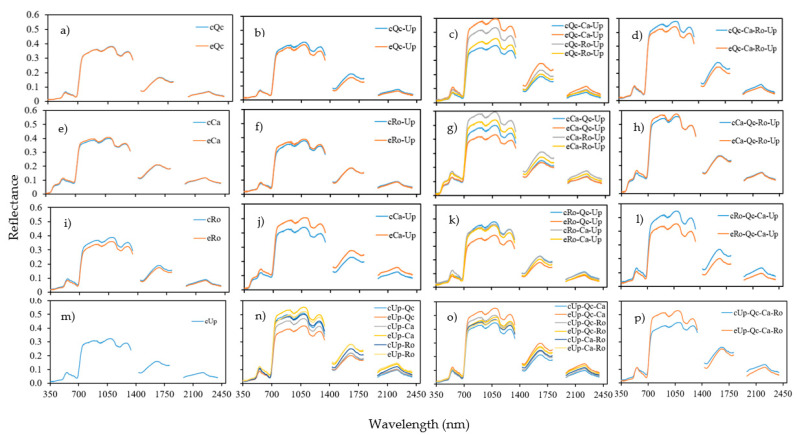
Spectral reflectance of assemblages in control (**c**) and exclusion (**e**). (**a**–**d**) *Quercus coccifera* (Qc); (**e**–**h**), *Cistus albidus* (Ca); (**i**–**l**) *Rosmarinus officinalis* (Ro); (**m**–**p**), *Ulex parviflorus* (Up) in monospecific. (**a**,**e**,**i**,**m**) Bispecific (**b**,**f**,**j**,**n**), trispecific (**c**,**g**,**k**,**o**) and tetraspecific (**d**,**h**,**l**,**p**). The first two letters of each assemblage indicate the measured species. Means of *n* = 27.

**Figure 7 plants-12-00879-f007:**
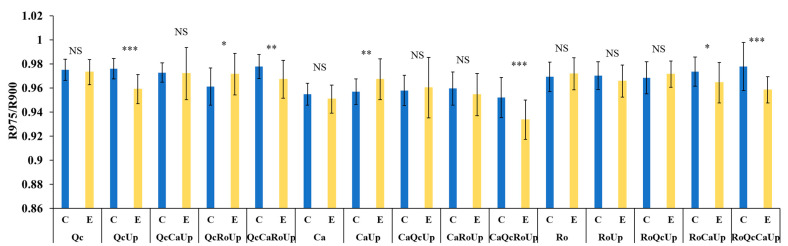
Histograms of drought index (R975/R900) of assemblages in control (C) and rain-exclusion (E) conditions. *Quercus coccifera* (Qc), *Cistus albidus* (Ca) and *Rosmarinus officinalis* (Ro) and *Ulex parviflorus* (Up). The first two letters of each assemblage indicate the species measured. NS, not significant; * *p* < 0.05; ** *p* < 0.01; *** *p* < 0.001. *n* = 27.

**Figure 8 plants-12-00879-f008:**
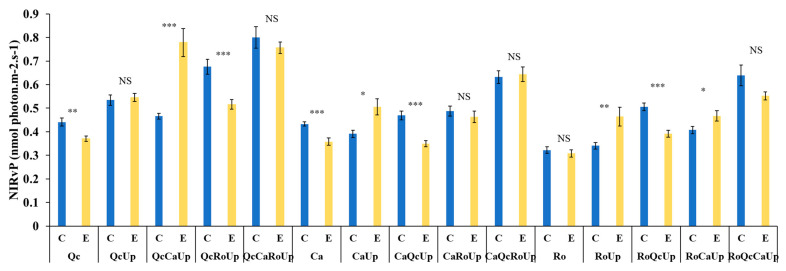
Histograms of photosynthesis and fluorescence index (NIRvP) of assemblages in control (C) and rain-exclusion (E) conditions. *Quercus coccifera* (Qc), *Cistus albidus* (Ca) and *Rosmarinus officinalis* (Ro) and *Ulex parviflorus* (Up). The first two letters of each assemblage indicate the species measured. NS, not significant; * *p* < 0.05; ** *p* < 0.01; *** *p* < 0.001. *n* = 27.

**Figure 9 plants-12-00879-f009:**
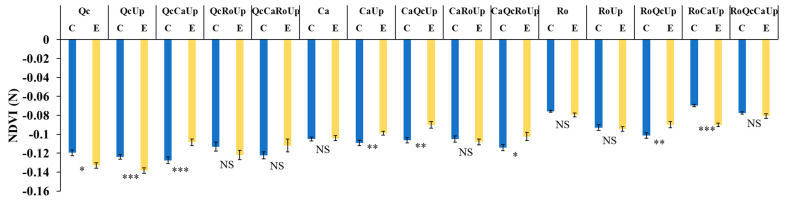
Histograms of nitrogen-content index, NDVI(N) of assemblages in control (C) and rain-exclusion (E) conditions. *Quercus coccifera* (Qc), *Cistus albidus* (Ca) and *Rosmarinus officinalis* (Ro) and *Ulex parviflorus* (Up). The first two letters of each assemblage indicate the species measured. NS, not significant; * *p* < 0.05; ** *p* < 0.01; *** *p* < 0.001. *n* = 27.

**Figure 10 plants-12-00879-f010:**
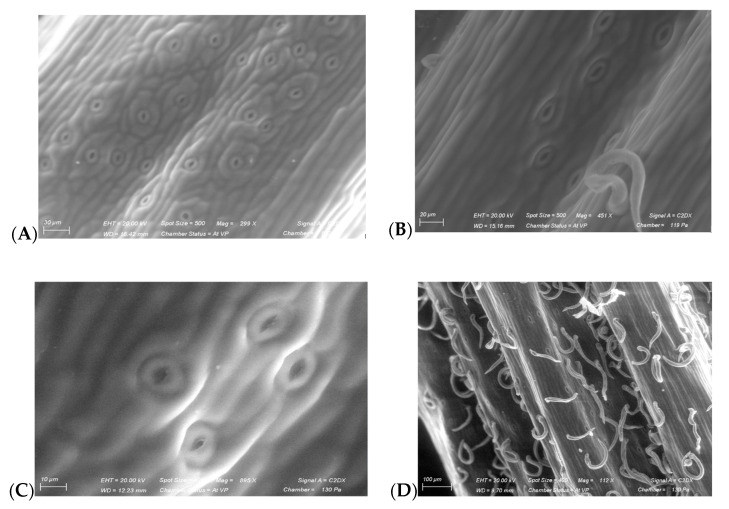
*Ulex parviflorus* stomata on young spine (**A**) and stem (**B**). Stomata and trichomes from mature spine (**C**) and stem (**D**). Scanning electronic microscopy (SEM).

**Table 1 plants-12-00879-t001:** Selected vegetation indices in the literature.

Index	Abbreviation	Formula	Reference
Normalized Phaeophytinization index	NPQI	(R415 − R435)/(R415 + R435)	[19]
Normalized Difference Vegetation Index (Nitrogen)	NDVI (N)	R565 − 708/R565 + R708	[20]
Normalized Difference Vegetation Index (Nitrogen)	NDVI (N) 2	(R730 − R759)/(R730 + R759)	[20]
Normalized Difference Vegetation Index (Nitrogen)	NDVI (N) 3	(R717 − R770)/(R717 + R770)	[20]
Normalized Difference Vegetation Index (Nitrogen)	NDVI (N) 4	(R720 − R839)/(R720 + R839)	[20]
Water Index	WI	R900/R970	[21]
Normalized Difference Vegetation Index	NDVI	(R800 − R680)/(R800 + R680)	[22]
Photochemical Reflectance Index	PRI	(R531 − R570)/(R531 + 570)	[23]
Solar Induce Florescence A proxy	SIFa	R740/R630	[24]
Solar Induce Florescence B proxy	SIFb	R685/R850	[24]
Enhansed vegetation index	EVI	2.5 × [(NIRref − REDref)/(NIRref + 6 × REDref − 7.5 × BLUEref)] + 1	[25]
Normalized Difference Index	NDI	(R850 − R710)/(R850 + R680)	[26]
Normalized Difference Red Edge	NDRE	(R800 − R720)/(R800 + R720)	[19]
Red Edge Index	RE	R740/R720	[27]
Red Edge Chlorophyll Index 1	CI1	(R800/R740) − 1	[28]
Modified Chlorophyll Absorption in Reflectance Index	MCARI	[(R700 − R1510) − 0.2(R700 − R550)] × (R700/R1510)	[29]
Nitrogen Planar Domain Index 2	NPDI2	(CI2 − CI2MIN)/(CI2MAX − CI2MIN)	[30]
Red Edge Chlorophyll Index 2	CI2	(R740/R550) − 2	[30]

## Data Availability

The data presented in this study are openly available in repository Zenedo at doi, 10.5281/zenodo.7641528.

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
