# Peer review of "The Role of Gorse (Ulex parviflorus Pourr. Scrubs) in a Mediterranean Shrubland Undergoing Climate Change: Approach by Hyperspectral Measurements"

_plants, 2023, doi:10.3390/plants12040879_

Round 1

Reviewer 1 Report

The paper deals with an interesting topic that is the better understanding of the ecological relationships between different species of plants in a community structure. This is relevant for the journal.

The underlining concept is that optical properties of Goarse change depending on the specific composition of the community. While the objectives of the study are clearly outlined at the end of the introduction, the abstract it is unclear, and I suggest to revise its structure. On the contrary the structure of the discussioni is satisfactory.

I really do not understand the need of including par 3.4.3 in the results: If differences in morphological structure are investigated depending on the community composition it makes sense adding this part, otherwise, and it is the case, it is redundant for the purpose of the study

Author Response

Reviewer 1

Comments and Suggestions for Authors

The paper deals with an interesting topic that is the better understanding of the ecological relationships between different species of plants in a community structure. This is relevant for the journal.

The underlining concept is that optical properties of Goarse change depending on the specific composition of the community. While the objectives of the study are clearly outlined at the end of the introduction, the abstract it is unclear, and I suggest to revise its structure. On the contrary the structure of the discussioni is satisfactory.

The abstract has been revised in accordance with the comments.

I really do not understand the need of including par 3.4.3 in the results: If differences in morphological structure are investigated depending on the community composition it makes sense adding this part, otherwise, and it is the case, it is redundant for the purpose of the study

Because of the significant mortality of gorse on the site, we tried to analyze its morphoanatomical structures in SEM. For clarity, these results are discussed with literature data see line 316-317.

Thank you for the comments and suggestion.

Regards

Reviewer 2 Report

Paper entitled “The Role of Gorse (Ulex parviflorus Pourr. Scrubs) In a Mediterranean Shrubland Submitted to Climate Change: Approach by Hyperspectral Measurements” addresses the effect of gorse on three typical species of a Mediterranean shrubland using hyperspectral information and vegetation indices. The analysis methodology was carefully established, selecting a total of 93 plots with different arrangements with rain exclusion and control devices and considering several types of plant combinations. The work has demonstrated that hyperspectral spectrometry has proven to be a promising nondestructive method in observing the neighbor effect in plant-plant interactions.

The paper is fine, well organized, and easy to follow. Many subsections, figures are included for the sake of clarity. The introduction section briefly contextualizes the work performed and clearly enumerates the goals of the research work.

Section 2.5 has to be improved. An introductory paragraph is missing. Also, some vegetation indices are explained at the beginning of the section but are not included in the table. For the sake of clarity, I suggest having all of them in a single place. In my opinion, a more detailed review of the state-of-the-art of vegetation indices would be desirable. In this context, in Table 1 authors list the vegetation indices selected. As there are hundreds of vegetation indices (for example https://www.indexdatabase.de/ or https://www.l3harrisgeospatial.com/docs/vegetationindices.html), authors have to provide some justification about the rational to include them. In fact, a common vegetation index in semi-arid regions with shrubs is MSAVI2 and it has not been included. Just a couple of references could be: https://rangeland.borujerd.iau.ir/article_520585_f759047bd5d383ff92002dcaaf7e3fda.pdf

https://www.tandfonline.com/doi/full/10.1080/15481603.2018.1502910

Figure 5 is blurred, and some letters are cut. It has to be improved.

In Figures 5 and 6, use A, B,.. instead of a), b),... to be  consistent with the caption of each figure and of Figure 4, as well.

As the Discussion section is long, I would add a Conclusions section including some of the existing information of section 4.

Fig. 2 shows the PLSDA of the reflected spectra. I imagine that such values depend on the specific season/date (phenological stage) when spectroradiometer measurements were taken. Authors should have to provide some information about this aspect. I don´t know if the figure could be improved incorporating, as well, the standard deviation of the spectral signature for each species. I let authors to decide because maybe too much information for the 4 species in a single figure could add confusion.

The reflectance was obtained between 350 and 2500 nm using a portable spectroradiometer

I suggest authors to carefully read the paper to include some missing “,” (just as an example, in line 40, comma is missing after “flammability” and between “thus”) and correct some typos (ie. Scare in line 59, was confirm in 159, etc.).

Author Response

Reviewer 2

Comments and Suggestions for Authors

Paper entitled “The Role of Gorse (Ulex parviflorus Pourr. Scrubs) In a Mediterranean Shrubland Submitted to Climate Change: Approach by Hyperspectral Measurements” addresses the effect of gorse on three typical species of a Mediterranean shrubland using hyperspectral information and vegetation indices. The analysis methodology was carefully established, selecting a total of 93 plots with different arrangements with rain exclusion and control devices and considering several types of plant combinations. The work has demonstrated that hyperspectral spectrometry has proven to be a promising nondestructive method in observing the neighbor effect in plant-plant interactions.

The paper is fine, well organized, and easy to follow. Many subsections, figures are included for the sake of clarity. The introduction section briefly contextualizes the work performed and clearly enumerates the goals of the research work.

Section 2.5 has to be improved. An introductory paragraph is missing. Also, some vegetation indices are explained at the beginning of the section but are not included in the table. For the sake of clarity, I suggest having all of them in a single place. In my opinion, a more detailed review of the state-of-the-art of vegetation indices would be desirable. In this context, in Table 1 authors list the vegetation indices selected. As there are hundreds of vegetation indices (for example https://www.indexdatabase.de/ or https://www.l3harrisgeospatial.com/docs/vegetationindices.html), authors have to provide some justification about the rational to include them. In fact, a common vegetation index in semi-arid regions with shrubs is MSAVI2 and it has not been included. Just a couple of references could be: https://rangeland.borujerd.iau.ir/article_520585_f759047bd5d383ff92002dcaaf7e3fda.pdf

https://www.tandfonline.com/doi/full/10.1080/15481603.2018.1502910

The section has been improved with an introduction and a citation of a publication from the reference above-mentioned. We agree with the importance of the VI MSAV12. However, our measurements were carried out at canopy level under the same synoptic conditions and soil homogeneity.

Figure 5 is blurred, and some letters are cut. It has to be improved.

The figure has been improved.

In Figures 5 and 6, use A, B,.. instead of a), b),... to be  consistent with the caption of each figure and of Figure 4, as well.

This is done.

As the Discussion section is long, I would add a Conclusions section including some of the existing information of section 4.

A conclusion section is done.

Fig. 2 shows the PLSDA of the reflected spectra. I imagine that such values depend on the specific season/date (phenological stage) when spectroradiometer measurements were taken. Authors should have to provide some information about this aspect. I don´t know if the figure could be improved incorporating, as well, the standard deviation of the spectral signature for each species. I let authors to decide because maybe too much information for the 4 species in a single figure could add confusion.

The date of the measurements is shown. We also added the standard errors in the figure 3

The reflectance was obtained between 350 and 2500 nm using a portable spectroradiometer

I suggest authors to carefully read the paper to include some missing “,” (just as an example, in line 40, comma is missing after “flammability” and between “thus”) and correct some typos (ie. Scare in line 59, was confirm in 159, etc.).

This is checked.

Thank you for the comments and suggestion.

 Regards
